# Recycled Waste Leaf Litter Pots Exhibit Excellent Biodegradability: An Experimental Analysis

**Daegeun Ko [1], Haegeun Chung [1], Jongbae Park [2], Hyungwoo Kim [3], Eunseo Kang [3], Songhee Lee [4] and Tae Kyung Yoon [3],\***

1 Department of Environmental Engineering, Konkuk University, Seoul 05029, Republic of Korea; poohgeuny@gmail.com (D.K.); hchung@konkuk.ac.kr (H.C.)
2 Flowerlump, Wonju 26339, Republic of Korea; zozop0118@flowerlump.com
3 Department of Forest Science, Sangji University, Wonju 26339, Republic of Korea; khw990222@naver.com (H.K.); eee1035@naver.com (E.K.)
4 Department of Smart Farm, Kyungmin University, Uijeongbu 11618, Republic of Korea; sh10241214@gmail.com
\* Correspondence: yoon.ecology@gmail.com

**Abstract:** The growth of the gardening kit market could result in the increased wasting of nursery pots, which are usually made of plastic. Replacing these pots with biodegradable pots made from green waste could have benefits for climate mitigation, the circular economy, and the greenness of gardening. To address this, we introduce a prototype recycled waste leaf litter (RWLL) nursery pot. Via an incubation experiment over 90 d, we examined their biodegradability and effects on microbial enzyme activity and inorganic nitrogen concentration, comparing them with commercially available biodegradable pots, namely peat–paper mixture pots (also known as Jiffypots®) and coco-coir pots. The effects of pot thickness were tested. Based on mass loss during incubation and on soil $CO_2$ efflux, the RWLL pots exhibited excellent biodegradability, regardless of their thickness, with decomposition rates and soil $CO_2$ efflux 1.5–6 times greater than other biodegradable pots. Biodegradability, extracellular enzyme activity, and soil inorganic nitrogen content were not affected by RWLL pot thickness or by the presence or absence of a plant in the soil. Unlike in natural ecosystems, leaf litter is treated as waste in urban green spaces, and its decomposition into soil organic matter is prevented. Creating plant pots from leaf litter enhances soil quality, reduces atmospheric carbon emissions, and satisfies the desire of gardeners for greenness.

**Keywords:** biodegradable pot; climate mitigation; coco coir; extracellular enzyme acidity; Jiffypot®; green waste management



## 1. Introduction

Gardeners who try to grow plants indoors or in the garden often fail at this because of improper plant selection or a lack of experience in plant care [1]. Gardening kits, which include seedlings, pots, soil, and gravel, are an attractive solution for beginner gardeners. Gardening kits also function as symbols of greenness, ecological soundness, and sustainability. Gardening kits are in demand for extracurricular activities in childhood education, as well as in horticulture therapy, public programs, and by a variety of individuals [2,3]. Their importance as a tool against social distancing increased during the COVID-19 pandemic [4]. For example, the pandemic introduced 18.3 million new gardeners in the USA, according to the 2021 National Gardening Survey [5], and increased the gardening retail sales revenue from approximately 5% for 2014–2019 to 8.79% in 2020, according to the Advance Monthly Retail Trade Survey of the United States Census Bureau [6]. In South Korea, the per capita flower consumption increased by 6.1% in 2021, which contrasts with the steady annual decline of 3.5% observed for 2005–2020 [7].

Nonetheless, gardening kits that are sold online incur the unexpected problem of creating packaging waste [8]. In particular, nursery plant pots, which are usually made from plastic, are disposed of immediately after the seedling has been transplanted into a plant pot, contributing to waste and increasing the carbon footprint. To address this situation, the amount of waste associated with gardening kits should be minimized [9].

Biodegradable pots are a sustainable and environmentally friendly alternative to plastic pots because they reduce plastic waste and environmental pollution [10]. Despite the energy requirements and costs involved in their manufacturing, they can be considered eco-friendly both because they utilize waste resources and because they are compostable. Furthermore, the seedling does not need to be removed from the pot for transplanting.

Biodegradable pots have been created from various types of organic waste (Table 1), including tomato and hemp fiber [11], textile and paper waste [12], cattle manure and wood waste [13], cassava starch containing various agro-industrial residues [14], banana peels combined with biomaterials [15], paddy straw and starch [16], and agro-industrial wastes combined with byproducts [17]. Despite the considerable public interest in biodegradable pots, few studies have examined their biodegradability.

**Table 1.** Prior studies examining biodegradable nursery pots.

| Source | Pot Materials | Measured Variables | Key Findings |
|---|---|---|---|
| Schettini et al. [11] | Recycled residues of tomato and hemp fibers | - Physical properties: density, porosity, water uptake, water absorption<br>- Scanning electron microscopy (SEM)<br>- Mechanical properties: flexural and tensile strength<br>- $CO_2$ production<br>- Plant seedling height and root development | The pots degraded completely within 16 d of transplanting, allowing the passage of the roots through the container walls. |
| Juanga-Labayen and Yuan [12] | Textile waste (cotton and polycotton) blended with paper substrates (newspaper and corrugated cardboard) | - Mechanical properties: tensile and bending strength<br>- Degradability of pots: anaerobic assay–specific $CH_4$ yield, biogas yield, % COD reduction, % volatile solids reduction; % weight loss of soil buried pots<br>- Seed germination | The pots degraded faster than Jiffypots® during a 120 d soil burial test. |
| Manafi-Dastjerdi et al. [13] | Cattle manure and sawdust with natural binders (cornstarch, sheep's wool) | - Physical properties: water absorption, thickness swelling<br>- Mechanical properties: rupture load, internal bonding strength<br>- Decomposition periods<br>- Plant root and stem length | Pots containing sheep's wool decomposed in 33 d, while the control pots decomposed in 51 d. |
| Ferreira et al. [14] | Cassava starch containing agro-industrial residues (sugarcane bagasse, cornhusk, malt bagasse, and orange bagasse) | - Physical properties: thickness, density, water absorption capacity<br>- Fourier-transform infrared spectroscopy (FTIR), SEM<br>- Mechanical properties: tensile strength, elongation<br>- Mass loss of buried tray | After 60 d, only those trays containing 20–30% orange bagasse were completely degraded. |

**Table 1.** *Cont.*

| Source | Pot Materials | Measured Variables | Key Findings |
|---|---|---|---|
| Mohd Rafee et al. [15] | Biomaterials (tapioca starch, water, vinegar, and glycerol) and banana peels | - Decomposition rate: weight loss percentage of pots | Weight loss during decomposition varied significantly between the different types of biodegradable pots, and was affected by the ratio of banana peels. |
| Pratibha et al. [16] | Paddy straw as the filler (untreated, alkali-treated, or alkali-treated and autoclaved) with six different biocomposites: corn starch (native or cross-linked using boric acid) as the matrix, and glycerol as a plasticizer | - Physical properties: % water uptake, % disintegration in aqueous medium, porosity %, and density<br>- Mechanical properties: tensile strengths<br>- SEM<br>- Macro- and micro-nutrients of different biocomposites<br>- Antimicrobial activity<br>- Biodegradation: $CO_2$ emission, % weight loss, cultivable method for soil bacteria and fungi, and FDA hydrolysis assay<br>- Plant growth, visible pot degradation, and root penetration | Cucumber plants along with their biodegradable pots were transplanted into fields. Both types of pots disintegrated within 10–20 d after transplantation, allowing the passage of the roots through the container walls. |
| Fuentes et al. [17] | Agro-industrial wastes and byproducts: gelatin, wheat–waste flour, corn-waste flour, cellulose paper, sunflower seed husks, rice husks, and yerba mate waste | - C/N ratio of biocomposites<br>- Physical properties: density, solubility, water absorption<br>- Mechanical properties: tensile and flexural strength<br>- Mass loss of buried pots<br>- Plant growth rate | The gelatin-based biocomposite pot showed the highest decomposition rate (62%) while the others showed rates <28% during a 24 d experiment. |

In urban environments, green spaces are beneficial for maintaining the climate (e.g., local temperature, $CO_2$, and hydrology), supporting biodiversity, providing recreational opportunities, enhancing landscape aesthetics, and thus improving quality of life [18–20]. In these environments, natural processes (including autumn leaf fall and tree mortality) and maintenance practices (such as pruning, weeding, and mowing) generate significant amounts of green waste, comprising woody debris, fallen leaf litter, pruned branches, and weeds [21–23]. Attempts have been made to recycle green waste via composting [24], as a source of bioenergy [25], as biochar [26], and for material utilization [27] in terms of the circular economy [28,29]. Biodegradable pots have been constructed from waste plant materials such as coffee and coco coir. Recycling leaf litter to create biodegradable pots, therefore, presents an attractive way of utilizing urban green waste.

To address this need, we developed a recycled waste leaf litter (RWLL) nursery pot for indoor gardening using gardening kits (Figure 1). We aimed to investigate the biodegradability of RWLL pots in comparison with other commercially available biodegradable pots and to test their effects on soil microbial activity and nutrient levels.

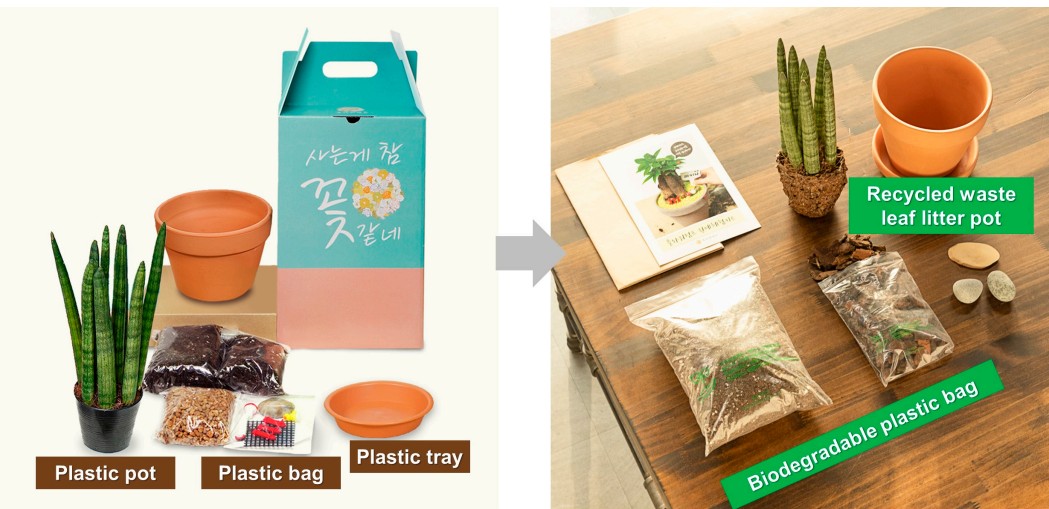

**Figure 1.** Development of a recycled waste leaf litter pot for environmentally friendly gardening kits, minimizing the use of disposable products.

## 2. Materials and Methods

### 2.1. Experimental Design

We designed the incubation experiment to address three research objectives (Figure 2). First, we compared the performance of RWLL pots with that of commercially available biodegradable pots (see research question [RQ] 1 in Figure 2). To investigate this, peat–paper and coco-coir pots were tested together with an RWLL pot. Second, we tested whether RWLL pot thickness affected biodegradability, soil microbial activity, and nutrient content (see RQ2 in Figure 2). To examine this, two pot thicknesses were compared. Third, we examined whether plant roots affected pot decomposition, soil microbial activity, and nutrient content (see RQ3 in Figure 2). To address this, we incubated the pots with soil only or with soil and plants.

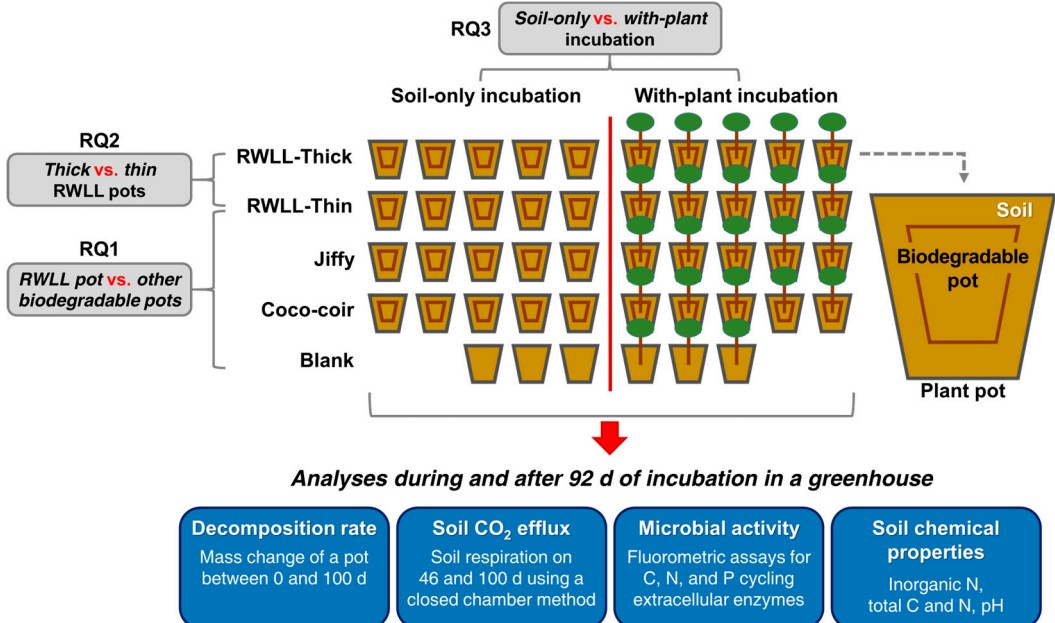

**Figure 2.** Design of and measures taken during the incubation experiment for testing the biodegradability of recycled waste leaf litter (RWLL) pots. RQ1–3 refers to the three research questions described in Section 2.1. The blank pots comprise controls, that is, soil in clay pots. In all other cases, soil was placed inside the biodegradable nursery pot located within a plant pot made of clay.

Five replicates were used for each set of incubation experiments. All of the pot types were subjected to soil-only or soil-with-plant incubation. Three blanks (soil but no biodegradable pot) were subjected to the same incubation treatments (Figure 2).

### 2.2. Soil and Plant Preparation

The soil used in our experiment was commercially available JiffySubstrates® (Jiffy Products International BV, Zwijndrecht, The Netherlands) in a peat-based growth medium. The pH, total carbon, and nitrogen content were 5.9, 43.77%, and 0.93%, respectively. The dwarf umbrella tree (*Heptapleurum arboricola* Hayata), an evergreen shrub that is commonly grown as both a houseplant and a garden plant, was used in this study.

### 2.3. Greenhouse Incubation

Fallen leaf litter, mostly from broad-leaved trees, was collected from local greenspaces in Wonju City, South Korea. To produce the RWLL pots, the leaves were washed, sterilized, crushed, mixed with starch, molded, and oven-dried for several days before the incubation experiment. Commercially available Jiffypots® (Jiffy Products International BV, Zwijndrecht, The Netherlands) made from peat–paper mixture [12,30] were obtained, and coco-coir pots were prepared.

The biodegradable plant-material pots were incubated for 92 d (22 July to 21 October 2022) in a local greenhouse. Each biodegradable pot was placed inside a clay pot and was filled with 330 g of soil for soil-only incubation or 280 g of soil for with-plant incubation. For the with-plant incubation, a seedling of a dwarf umbrella tree (ca. 20 cm in height) was planted in each pot. The daytime temperature and humidity in the greenhouse were maintained at 25–30 °C and 30–80%, respectively. Sufficient water was provided via regular watering, as per standard horticultural practices.

### 2.4. Field and Laboratory Measurements

Soil $CO_2$ efflux (i.e., soil respiration) was determined at 46 and 92 d after the start of incubation as an indicator of microbial activity during pot decomposition. A closed chamber ($1254 \text{ cm}^2$) was placed on the soil surface of each clay pot, and a portable infrared gas analyzer (GMP252, Vaisala, Sweden) was used to detect an increase in $CO_2$ concentration inside the chamber for at least 5 min. The soil $CO_2$ efflux was measured only for the soil-only pots because of the difficulty in separating the $CO_2$ flux of the microbial pot's decomposition from that of plant metabolism, including photosynthesis and autotrophic respiration.

Pot decomposition rates were determined based on mass loss during incubation. Before incubation, the biodegradable pots were oven-dried at 65 °C for 48 h, and their constant weight was recorded. At the end of the incubation period (after 92 d), the incubated pots were excavated, the soil was carefully removed, the pots were oven-dried, and the weight was measured again. The change in the mass of the pot during incubation was assumed to indicate decomposition.

At the end of the incubation period, two sets of soil samples were collected from each pot. The sample for microbial analyses was stored in a sterilized polyethylene sample bag and was frozen at −20 °C without air-drying. The other sample was air-dried and pretreated.

Fluorometric assays using methylumbelliferone (MUB)-linked substrates were performed to determine the microbial community's metabolism. We determined the activity of five extracellular enzymes (cellobiohydrolase [CBH], β-1,4-glucosidase, β-1,4-xylosidase, β-1,4-N-acetylglucosaminidase, and acid phosphatase [AP]) that are involved in C, N, and P cycling in the soil [31,32]. Briefly, the soil slurry made using sodium acetate buffer was transferred to a 96-well black microplate. Plates containing all five enzymes were incubated at room temperature for 2 h. Fluorescence was measured using a Synergy HT Multi-Mode Microplate Reader (BioTek Instruments Inc., Winooski, VT, USA); excitation energy was set at 360 nm, and the emission energy was measured at 460 nm. Enzyme activity was expressed as nmol 4-MUB $g^{-1}$ $h^{-1}$.

Colorimetric assays were performed to determine the inorganic N content. Five grams of soil was extracted using 2 M of potassium chloride (KCl). The KCl extracts were transferred to a 96-well clear microplate, and the ammonium ($NH_4^+$) and nitrate ($NO_3^-$) concentrations were determined colorimetrically using the salicylate–nitroprusside and vanadium(III) reduction methods, respectively [33–35]. Absorbance was measured using a Synergy HT Multi-Mode Microplate Reader, with absorbance set at 650 nm for $NH_4^+$ and 540 nm for $NO_3^-$.

*2.5. Statistical Analyses*

Two-way ANOVA was performed to test the effects of using different types of biodegradable pots and of soil-only vs. with-plant incubation. The interaction terms were not significant in the preliminary analysis and were, therefore, excluded from ANOVA testing. For significant effects, Tukey's honest significant difference post hoc testing was performed. Data analysis was performed using R 4.2.2 [36].

**3. Results**

*3.1. Pot Biodegradability*

Decomposition rates differed significantly among the pots ($p < 0.001$) but not between the with-plant and soil-only treatments ($p = 0.21$) (Figure 3). RWLL pot thickness did not affect the decomposition rate. Under soil-only incubation, both the thick and thin RWLL pots decomposed by ca. 70%, which is much more than the other pot types (Jiffy, 23%; coco-coir, 12%). For the thick RWLL pots, the decomposition rate was 16% lower in the with-plant group than in the soil-only group (Figure 3).

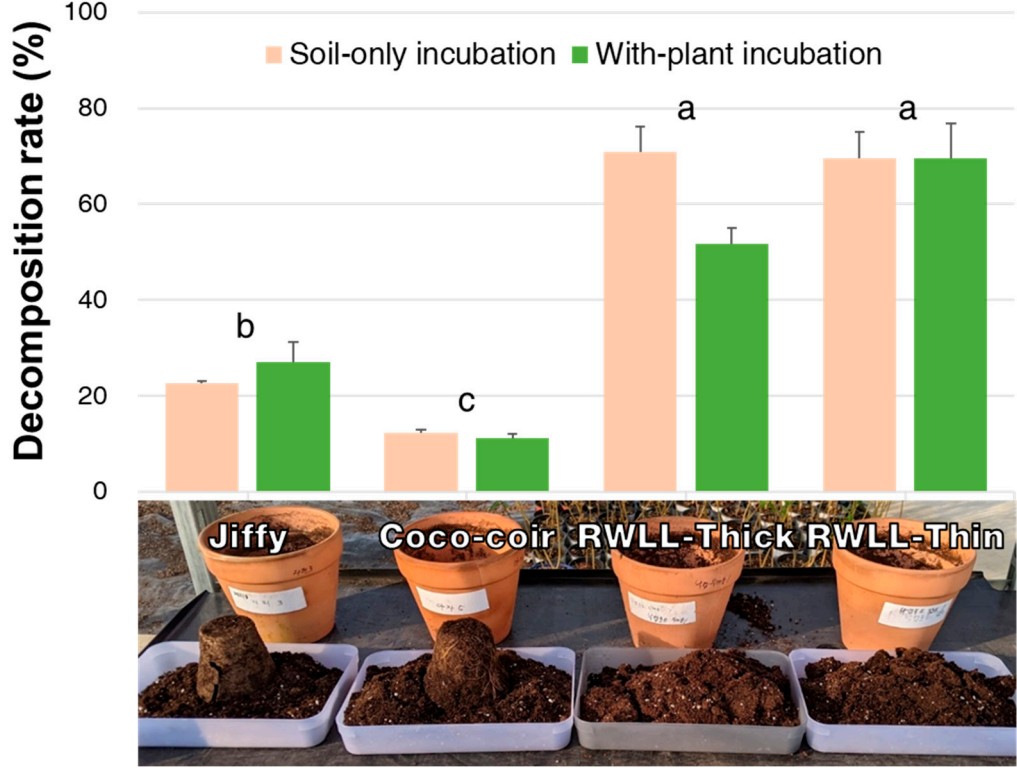

**Figure 3.** Decomposition rates of biodegradable pots incubated in a greenhouse for 92 d. Error bars indicate the standard error of the mean ($N = 5$). Differences in decomposition rate were significant among the biodegradable pots ($p < 0.001$), not between the soil-only and with-plant treatments ($p = 0.21$; two-way ANOVA). Lowercase letters above the bars identify groups of pot types that differed significantly (Tukey's HSD; $p < 0.05$).

Based on a visual inspection of the extracted pots at the end of the 92 d incubation period, the Jiffy and coco-coir pots retained their original shape. By contrast, the RWLL pots had decomposed considerably and were physicochemically mixed with the soil, making it difficult to retain their original shape (Figure 3).

Soil $CO_2$ efflux due to microbial metabolism differed significantly among the biodegradable pots at day 46 of incubation ($p < 0.001$) but not at day 92 ($p = 0.14$) (Figure 4). At day 46, the $CO_2$ efflux (in mg C $m^{-2}$ $h^{-1}$) was 4.5 for the blanks, 9.1 for Jiffy pots, 25.8 for coco-coir pots, 56.3 for thick RWLL pots, and 37.1 for thin RWLL pots. The thick and thin RWLL pots emitted 1.5–6.0 times more $CO_2$ than the Jiffy and coco-coir pots. However, the overall $CO_2$ efflux and the differences among the biodegradable pots in $CO_2$ efflux were limited at day 90 of incubation. At day 92, $CO_2$ efflux was 3.3 for the blanks, 3.3 for Jiffy pots, 8.2 for coco-coir pots, 13.3 for thick RWLL pots, and −2.8 for thin RWLL pots (Figure 4).

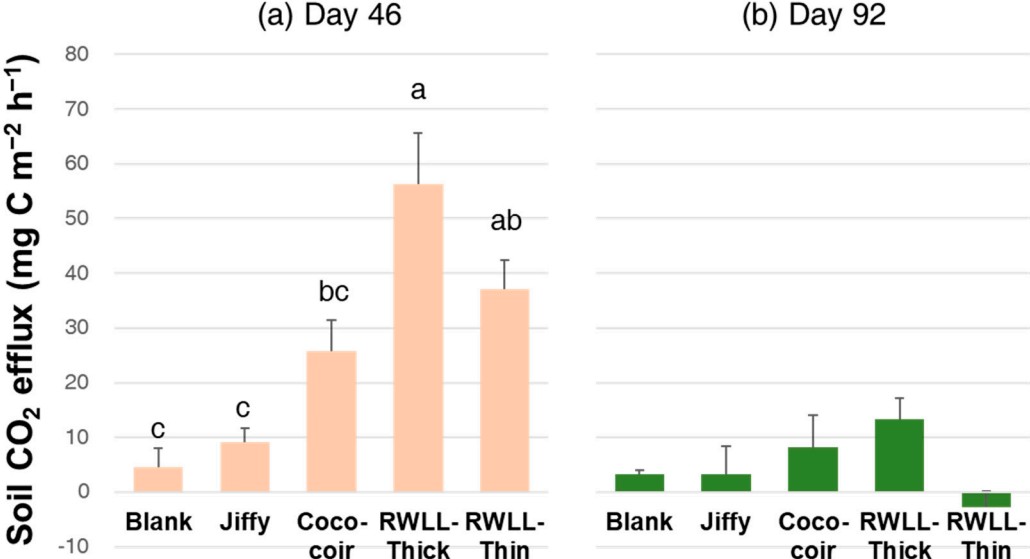

**Figure 4.** Soil $CO_2$ efflux from the soil in biodegradable pots on days 46 and 92 of incubation in a greenhouse. Error bars indicate the standard error of the mean ($N = 5$). Lowercase letters above the bars identify groups of pot types that differed significantly (Tukey's HSD; $p < 0.05$).

### 3.2. Extracellular Enzyme Activity and Soil Chemical Properties

Among the five soil extracellular enzymes tested, CBH and AP exhibited significantly different activity among the pot types ($p < 0.01$) (Table 2) for the soil-only and with-plant incubation treatments, respectively. Compared to the blanks and thick RWLL pots, the activity of CBH was significantly higher in the coco-coir pots ($p < 0.01$). In the with-plant treatment, AP activity was significantly higher in the Jiffy pots than in the coco-coir pots and thick and thin RWLL pots ($p < 0.01$).

The inorganic N content differed significantly among the biodegradable pots ($p < 0.01$) (Figure 5). The $NO_3^-$ content differed significantly between the pot types for both the soil-only and with-plant treatments, whereas the $NH_4^+$ content differed significantly among the pot types only for the with-plant treatment. For the blank, the soil inorganic N content was significantly higher under the soil-only treatment than under the with-plant treatment ($p < 0.05$), whereas for both the thick and thin RWLL pots, the soil inorganic N content was significantly higher under the with-plant treatment than under the soil-only treatment. For the blank and Jiffy pots, soil $NO_3^-$ content was significantly higher in the soil-only treatment than in the plant-only treatment ($p < 0.05$). The soil $NH_4^+$ content for both the thick and thin RWLL pots was significantly higher in the with-plant treatment than in the soil-only treatment. Soil pH, total carbon, and nitrogen content did not differ significantly among the biodegradable pots.

**Table 2.** Soil extracellular enzyme activity in the biodegradable pots incubated for 92 d in a greenhouse.

| | | Enzyme Activity (nmol 4-MUB g$^{-1}$ h$^{-1}$) | | | | |
|---|---|---|---|---|---|---|
| | | CBH | BG | BX | NAG | AP |
| Blank | Soil-only | 9.23 ± 1.83 b | 101.03 ± 13.30 | 4.45 ± 0.50 | 99.90 ± 9.98 | 217.61 ± 23.94 |
| | With-plant | 20.89 ± 4.87 | 94.73 ± 9.99 | 4.58 ± 0.63 | 71.31 ± 5.59 | 169.03 ± 21.37 ab |
| Jiffy | Soil-only | 12.73 ± 1.79 ab | 98.28 ± 5.76 | 5.27 ± 0.85 | 112.79 ± 8.28 | 235.88 ± 3.10 |
| | With-plant | 19.82 ± 2.59 | 113.86 ± 7.27 | 5.29 ± 0.74 | 97.45 ± 12.84 | 237.03 ± 20.58 a |
| Coco coir | Soil-only | 18.84 ± 1.28 a | 130.96 ± 8.89 | 4.72 ± 0.34 | 116.57 ± 9.37 | 231.13 ± 13.68 |
| | With-plant | 18.27 ± 3.78 | 111.87 ± 6.89 | 4.58 ± 0.31 | 83.85 ± 7.44 | 162.85 ± 7.09 b |
| RWLL-Thick | Soil-only | 11.88 ± 1.89 b | 116.19 ± 14.41 | 4.24 ± 0.70 | 128.31 ± 17.90 | 210.06 ± 17.94 |
| | With-plant | 17.87 ± 2.23 | 126.24 ± 18.68 | 4.71 ± 0.97 | 153.24 ± 42.17 | 160.98 ± 26.00 b |
| RWLL-Thin | Soil-only | 15.82 ± 1.20 ab | 114.56 ± 11.40 | 4.51 ± 0.43 | 113.60 ± 10.06 | 177.63 ± 12.52 |
| | With-plant | 12.78 ± 1.74 | 107.57 ± 16.63 | 3.41 ± 0.26 | 66.50 ± 7.27 | 109.79 ± 7.50 b |

Values are expressed as mean ± standard error ($N = 10$). The lowercase letters identify groups of pots that differed significantly (Tukey's HSD; $p < 0.05$). CBH: cellobiohydrolase; BG: β-1,4-glucosidase; BX: β-1,4-xylosidase; NAG: β-1,4-N-acetylglucosaminidase; AP: acid phosphatase.

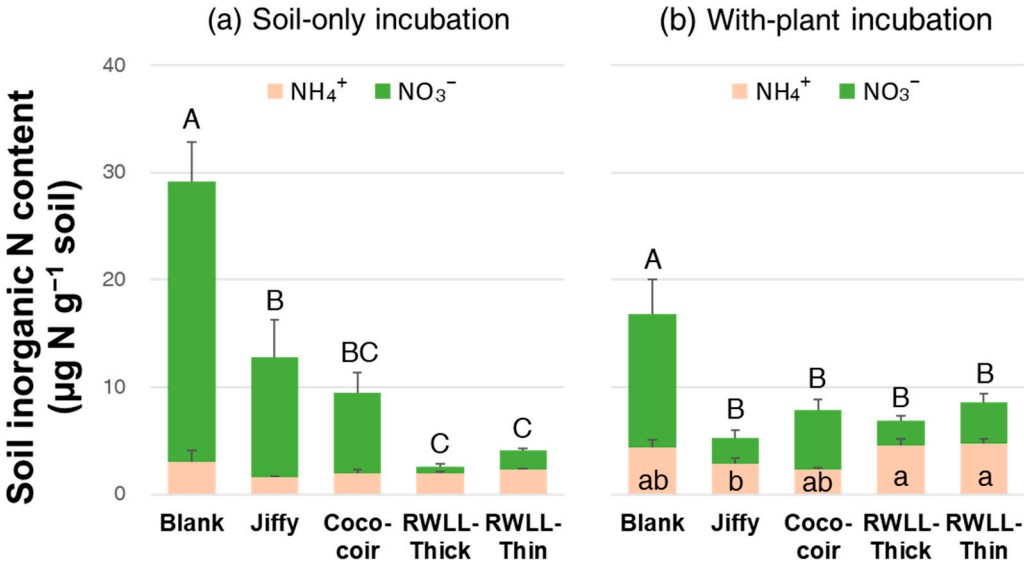

**Figure 5.** Soil inorganic nitrogen content following the greenhouse incubation of biodegradable pots for 92 d. Error bars indicate the standard error of the mean ($N = 10$). Lowercase and uppercase letters above the bars identify groups of pot types that differed significantly in the $NH_4^+$ and $NO_3^-$ content, respectively (Tukey's HSD; $p < 0.05$).

## 4. Discussion

### 4.1. Biodegradabilty, Microbial Activity, and Nitrogen Availability under Experimental Treatments

Our findings reveal that RWLL pots exhibited substantially better biodegradability than commercially available biodegradable pots. Regardless of thickness, the RWLL pots decomposed by >50% in soil with or without a plant present (Figure 3). They exhibited higher $CO_2$ efflux, supporting their excellent biodegradability (Figure 4). The activity of CBH for blanks, the Jiffy pots, and coco-coir pots showed a similar tendency for the soil $CO_2$ efflux at day 46, but not in the RWLL pots, and this suggests that C-cycling enzymes other than those analyzed in this study also mediate the decomposition process that leads to $CO_2$ efflux. Meanwhile, AP activity differed significantly among the different pots that had with-plant treatment. A significant decrease in soil $NO_3^-$ content in biodegradable pots was observed compared to the blanks (Figure 5), and this implies that nitrate may be absorbed by biodegradable pots. In particular, nitrate content was the lowest for both thick

and thin RWLL pots (Figure 5a), and thus, these pots may release a higher level of nitrogen during their degradation in soils while functioning as nursery pots.

RWLL pot thickness did not significantly affect biodegradability, soil microbial activity, or soil inorganic nitrogen content (Figures 3 and 5; Table 2). Only soil $CO_2$ efflux varied with thickness, probably because the thicker pots contained more decomposable material (weighing $1.5\times$ as much as the thin pots) rather than because of a qualitative difference in biodegradability.

The presence of a plant in the soil significantly affected the decomposition rate only for thick RWLL pots. In other words, for most of the pot types tested, plant root physiological activity did not promote decomposition. This implies that including a plant to simulate the actual growing condition is not mandatory for biodegradability testing. The plants reduced the soil nutrient content by absorbing the nutrients.

### 4.2. Application of RWLL Pots to Promote Climate-Smart Horticulture

Soil organic matter content, which reflects the ecological function and health of a given soil [37], depends largely on plant debris, including leaf litter [38]. In natural ecosystems, the presence of leaf litter and its transformation into soil organic matter fundamentally regulates the physicochemical and biological characteristics of soils and, consequently, ecosystem productivity [39,40]. In urban environments such as gardens, parks, and streets, the natural conversion of leaf litter to soil organic matter is obstructed [41], as leaf litter is usually collected and treated as waste material. Such clearing results in soil degradation by reducing organic matter input [42] and increases carbon emissions via waste disposal.

An approach that avoids removing waste leaf litter from natural or artificial environments could be beneficial for soils, climate, plants, and people. RWLL pots could (1) reduce carbon emissions by reducing waste disposal, (2) replace plastic pots, which have a higher carbon footprint [43,44] for nursery pot production, and (3) increase the recycling of waste materials, thereby supporting climate–smart horticulture [45]. Furthermore, RWLL pots can be manufactured, transported, and consumed locally, whereas other commercially available biodegradable pots (such as Jiffy and coco-coir pots) used in South Korea require cross-country shipping.

### 4.3. Differences of RWLL Pots Compared to Others

Biodegradable or compostable pots for use in horticulture, floriculture, and agriculture have been developed worldwide as an alternative to non-renewable petroleum-based plastic pots over the last few decades. The biodegradability of pots created from various organic wastes has been experimentally tested in previous studies (Table 1), which have incorporated materials such as wastes from agriculture, food, paper, textiles, and livestock industries. In contrast, our study utilized waste leaf litter: a green waste obtained from urban green spaces. Because the presence of urban green waste affects citizens physically and psychologically, the introduction of biodegradable RWLL pots may attract considerable public attention due to their green waste recycling potential.

Although the decomposition of RWLL pots within 92 d was confirmed, some limitations remain. Long-term decomposition patterns of biodegradable pots—which might be different from short-term patterns—were not assessed. Soil $CO_2$ efflux was only determined twice at approximately 50-d intervals, which excluded the evaluation of temporal patterns of early decomposition. Furthermore, the physical properties of RWLL pots (such as mechanical stability) were not determined. Further incubation experiments for biodegradable pots should address these limitations.

## 5. Conclusions

The production of plant nursery pots using waste leaf litter offers an environmentally friendly and economical alternative to commonly used plastic pots. RWLL nursery pots exhibited substantially better biodegradability than the other commercially available biodegradable pots that we tested. The RWLL pots exhibited high decomposition rates and $CO_2$ efflux. The inorganic N content, reflecting the nutrients available for plants and soil microbes, was significantly lower when a plant was present in the soil, mainly reflecting nutrient uptake by plants rather than by microbes. The activities of soil extracellular enzymes, including cellobiohydrolase and acid phosphatase, significantly differed among the biodegradable pots. Future research needs to determine the mechanical and horticultural performance of RWLL pots as well as the use of alternative waste materials for manufacturing them, such as pruned wood debris, mowed weeds, and processed waste compost from mushrooms. The manufacture and utilization of RWLL pots align entirely with a climate–smart strategy. These pots can satisfy the desire of gardeners to support green, ecological, and sustainable activities. RWLL pots, therefore, hold the promise of attracting strong social investment and commercial demand.

**Author Contributions:** Conceptualization, J.P. and T.K.Y.; methodology, H.C. and T.K.Y.; formal analysis, D.K.; investigation, H.K. and E.K.; resources, J.P. and S.L.; writing—original draft preparation, D.K. and T.K.Y.; writing—review and editing, H.C., S.L. and T.K.Y.; visualization, T.K.Y.; supervision, T.K.Y.; project administration, J.P. and T.K.Y.; funding acquisition, J.P. and T.K.Y. All authors have read and agreed to the published version of the manuscript.

**Funding:** This research was supported by the "Business support project for societal enterprises in the environmental sector" (B0080621001893) of the Korea Environmental Industry Association.

**Data Availability Statement:** The datasets generated and/or analyzed in the current study are available from the corresponding author upon reasonable request.

**Conflicts of Interest:** The authors declare no conflict of interest.

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
