# Peer review of "Recycled Waste Leaf Litter Pots Exhibit Excellent Biodegradability: An Experimental Analysis"

_horticulturae, doi:10.3390/horticulturae9090987_

Round 1
Reviewer 1 Report
1. In my opinion, the authors tried to find an alternative of plastic nursery pots that is quite environmentally friendly with an excellent biodegradability. 2. I think the authors performed the experiment well. However, they still need to justify their findings with other such types of studies. 3. The authors could add additional text in a separate sub-section, 4.3 at instance. 4. I think the methodology section is well written. However, the authors could add a 'Scheme diagram' or the overall experimental design of their study framework. 5. It would be good, if the authors improve the Conclusion section with mentioning the future use of the findings and the limitations of their study. 6. The text (i.e., size and bold) of the Figures and Tables are inconsistent. They should be in the same form.additional comments can be seen in the attachment.

The English of this manuscript is fine to me.
Reviewer 2 Report
Dear authors,
Your work is well-redacted and adequately carried out. However, some minor issues need revision:
1) In the introduction, could you provide some information about the gardening market, in order to justify the relevance of your work (statistics and references are welcome). Which is the worldwide trend of the gardening kit market (and production)?
2) Please, provide an adequate description of the soil that you used in the experiment. A description of chemical and physical is recommended. Probably, a sub-paragraph named "starting material" could be useful.
3) You have to choose a time of 100 days for your experiments. Could you justify this decision? In the conclusion paragraph, you could present further advances in the research that could include, for example, longer experiments (if necessary).
Reviewer 3 Report
The article is of considerable interest to specialists. A very original idea for assessing the effect of biodegradable materials in the soil on emissions.
In my opinion, it is necessary to add some soil characteristics that affect the rate of decomposition of biomaterial in the soil, such as: pH, organic carbon content, available forms of phosphorous and nitrogen for plants, granulometric composition of the soil, filtration rate.
It is advisable to specify these parameters before and after the experiment. I hope that these recommendations will help improve your manuscript.
Round 2
Reviewer 1 Report
Revised version of the manuscript is fine to me.